# Neurodynamics of Patients during a Dolphin-Assisted Therapy by Means of a Fractal Intraneural Analysis

**DOI:** 10.3390/brainsci10060403

**Published:** 2020-06-25

**Authors:** Oswaldo Morales Matamoros, Jesús Jaime Moreno Escobar, Ricardo Tejeida Padilla, Ixchel Lina Reyes

**Affiliations:** 1Escuela Superior de Ingeniería Mecánica y Eléctrica, Instituto Politécnico Nacional, 07340 Ciudad de México, Mexico; oswmm2001@yahoo.com (O.M.M.); megamikurayami6@gmail.com (I.L.R.); 2Escuela Superior de Turismo, Instituto Politécnico Nacional, 07630 Ciudad de México, Mexico; ricardotp75@hotmail.com

**Keywords:** neurodynamics, children disabilities, EEG, FFT-PSD, self-affine, TGAM1, dolphin-assisted therapy, BCI

## Abstract

The recent proliferation of sensor technology applications in therapies for children’s disabilities to promote positive behavior among such children has produced optimistic results in developing a variety of skills and abilities in them. Dolphin-Assisted Therapy (DAT) has also become a topic of public and research interest for these disorders’ intervention and treatment. This work exposes the development of a system that controls brain–computer interaction when a patient with different abilities undergoes a DAT. To develop the proposed system, TGAM1, i.e., ThinkGear-AM1 series of NeuroSky company, was used, connecting it to an isolated Bluetooth 4.0 communication protocol from a brackish and humid environment, and a Notch Filter was applied to reduce the input noise. In this way, at Definiti Ixtapa-Mexico facilities, we explored the behavior of three children with Infantile Spastic Cerebral Palsy (Experiment 1), as well as the behavior of Obsessive Compulsive Disorder and neurotypic children (Experiment 2). This was done applying the Power Spectrum Density (PSD) and the Self-Affine Analysis (SSA) from Electroencephalogram (EEG) biosignals. The EEG Raw data were time series showing the cerebral brain activity (voltage versus time) before and during DAT for the Experiment 1, and before, during DAT and after for the Experiment 2. Likewise, the EEW RAW data were recorded by the first frontopolar electrode (FP1) by means of an EEG biosensor TGAM1 Module. From the PSD we found that in all child patients a huge increment of brain activity during DAT regarding the before and after therapy periods around 376.28%. Moreover, from the SSA we found that the structure function of the all five child patients displayed an antipersistent behavior, characterized by $\sigma \propto {\left(\delta_{t}\right)}^{H}$, for before, during DAT and after. Nonetheless, we propose that one way to assess whether a DAT is being efficient to the child patients is to increase the during DAT time when the samples are collected, supposing the data fitting by a power law will raise the time, displaying a persistent behavior or positive correlations, until a crossover appears and the curve tends to be horizontal, pointing out that our system has reached a stationary state.

## 1. Introduction

Dolphins have had a therapeutic effect in the treatment of several human race conditions, both psychological and physical. Underwater communication sounds made by dolphins have played a role in this therapeutic effect. Dolphins appear to sense electrical fields from humans and attempt to communicate using the same frequencies. This has generated several suppositions: (a) Dolphin acoustic emissions, recorded in sea water, may bring about modifications in human brainwave activity, (b) dolphin interaction may give patients pain relief due to the increased release of hormones into the blood, (c) dolphin interaction may produce complex neurological, (d) stimulation helping relaxation, reducing stress levels and, thereby, strengthening the immune system, and (e) dolphin’s ultrasonic energy may cause significant cellular changes within the living tissue of the central nervous system [1].

Based on the above aspects, there has been developed an alternate treatment for people with diverse psychological and physical disabilities or disorders like Autism, Attention Deficit, Down Syndrome, Spastic Cerebral Palsy, and Obsessive Compulsive Disorder. This treatment is called Dolphin-Assisted Therapy (DAT). DAT is a growing area of interest to the general public, with reports of pain relief, extinction of depression and improved learning in children with different disabilities [2]. DAT aims to improve functioning through complementing and reinforcing existing therapies, rather than replacing them [1].

Regarding to Infantile Spastic Cerebral Palsy (ISCP) is a kind of Cerebral palsy (CP) that is defined as a group of non-hereditary neurological disorders that affect parts of the brain which control movement muscles in the body, is non-hereditary in nature and may not be detected immediately but appears in infancy or early childhood before 3 years old [3].

According to the extent, type and location of brain damage and/or abnormality, cerebral palsy are classified by type of movement disorder: Spastic (stiff muscles), athetoid (writing difficulties) or ataxic (poor balance and coordination), any of this may have paresis (weakness) or plegia (paralysis) of one or more extremities or sides of the body [3].

Spastic is the most common disorder among patients with CP, its types are: (i) Spastic hemiplegia/hemiparesis affects an arm or a hand and one side of the body, produces an abnormal curvature of the spine, seizures and usually patients have normal intelligence; (ii) spastic diplegia/diparesis presents muscle stiffness mostly in the legs, less severely affects the arms and face, could provoke clumsiness in the hands, hyperactive reflexes in the tendons of the legs, stiffness in the muscles of the legs (walking scissor) and normal intelligence and language skills; and (iii) spastic quadriplegia/quadriparesis is the severe form, as a result of malformations in brain or damage to a large part of children have intellectual disability, seizures difficult to control, severe stiffness in the extremities, very flexible neck, impossibility to walk and difficulties to speak [3].

The degree to which CP affects varies in each patient, in some it may require care for a lifetime or some even have the possibility of attending school since they have the necessary intelligence as the rest of the children his age, in this case children can achieve self-sufficiency and with help of physical therapies, medications and even surgery improve their motor skills and lead a full life [3].

In the case of Obsessive Compulsive Disorder (OCD), this disability is characterized by intrusive, troubling thoughts (obsessions), and repetitive, ritualistic behaviors (compulsions) which are time consuming, significantly impair functioning and/or cause distress. When an obsession occurs, it almost always corresponds with a massive increase in anxiety and distress. Common obsessions include contamination fears, worries about harm to self or others, the need for symmetry, exactness and order, religious/moralistic concerns, forbidden thoughts (e.g., sexual or aggressive), or a need to seek reassurance or confess. Common compulsions include: Cleaning/washing, checking, counting, repeating, straightening, routinized behaviors, confessing, praying, seeking reassurance, touching, tapping or rubbing, and avoidance. Unlike in adults, children need not view their symptoms as nonsensical to meet diagnostic criteria [4].

Younger children will not be able to recognize that their obsessions and compulsions are both unnecessary (e.g., they do not really need to wash their hands) and extreme (e.g., washing hands for 15–20 times is fine, but five minutes in scalding water is too much) in nature. In young children, compulsions often occur without the patient being able to report their obsessions, in the meantime adolescents are often able to report multiple obsessions and compulsions. Children and adolescents are also more likely to include family members in their rituals and can be highly demanding of adherence to rituals and rules, leading to disruptive and opposite behavior and even episodes of rage [4].

A way to estimate the DAT efficiency in children with different disabilities such as OCD is to observe changes in the children’ brainwave activity from Electroencephalogram (EEG) signals recorded before and during a DAT session. This EEG are yielded by means of the electroencephalography, which recording from scalp gives rise to the study of human cognitive activity through the measure of neurons’ electrical impulses.

Compared with other imaging techniques or behavioral observations such as the MagnetoEncephaloGraphy and the Magnetic Resonance Imaging, the electroencephalography has some advantages. For instance, its excellent time-resolution since several records can be taken in a second through multiple sensors which are cheaper, and its low-cost maintenance.

At the beginning, in the electroencephalography, the EEG data were often collected using of multiple electrodes and/or wired EEG headset to retain signal accuracy and spatial resolution, diminishing greatly the mobility and actual application pragmatism. To facilitate a mobile driver safety program, [5] demonstrated the effectiveness of wireless single-channel (electrode) EEG headset NeuroSky’s MindWave in distinguish user’s eyes conditions – open or closed. The MindWave Mobile is a Brain Computer Interface (BCI).

On the other hand, to develop an Artificial Intelligence system is necessary to include a physical environment model to operate the system [6]. Hence, we developed a model that measures the brain activity (physical feature) and the meaning of its behavioral changes. Therefore, this proposal can be considered as an Artificial Intelligence model since it incorporates the domain knowledge application (in this case, in Neuroscience) to be able to measure the DAT efficiency from the study of EEG biosignals.

Moreover, to study these EEG biosignals, it is necessary to use a BCI. In this way, BCI research programs have arisen due to more understanding of brain functions and powerful low-cost computer equipment. Under BCI, the users can communicate with environments and the external devices through their brain activities. In particular, people suffering from disabilities such as ISCP and OCD have growing number of demands and needs of such system. The goal of these BCI systems is to translate the neuro-physiological signals for controlling the external devices. For acquiring the brain dynamics, an EEG is relatively convenient, comfortable, and inexpensive [7].

Lastly, to get the rid out of economical loss, but still taking in account the improvement in the reliability, accuracy, latency and false alarm rate, [8] used TGAM1, that is, ThinkGear-AM1 series of NeuroSky company as a probe to detect the brain wave pattern from the user and designed filter and threshold algorithm to detect driver’s fatigues and drowsiness index with low cost and high reliability.

Accordingly to the above, this work has aimed to create a brain-computer system based on EEG biosignals. Thereby, it has been developed a system to analyze and measure the both OCD and neurotypic children brain activity underwent a DAT using the TGAM1 EEG signal sensor. Moreover, we applied the Power Spectrum Density and Self-Affine Analysis to study the EEG signals by means of BCI system, in order to estimate really changes in the children’ brainwave activity recorded before, during and after a DAT session.

It is important to highlight that this work made use of the system proposed by Moreno et al. in [9], where a quality of experience assessment based on Welch power density is proposed. This paper extends the mathematical tools used with the introduction of Self-Affine Analysis, for instance.

Hence, this paper is divided as follows: Section 2 describes the materials and methods to create the brain computer system for mesauring the DAT efficiency; Section 3 shows the results and discussion for the Experiment 1; Section 4 shows the results and discussion for the Experiment 2; in Section 5 are discussed our findings; and lastly in Section 6 the conclusions are presented.

## 2. Materials and Methods

### 2.1. Electroencephalography

The brain is made up of hundreds of thousands of cells called neurons, which interact with each other in a bioelectrical phenomenon called synapses. By means of impulses transmitted around the membrane, information is sent, helping the brain coordinate sensory, motor and cognitive functions. This electrical activity can be manifested by a lapse of tens to hundreds of milliseconds, sufficient for equipment such as EEG using non-invasive electrodes on the cranial surface to detect and record such activity.

Since human neuronal activity has a high-degree of complexity in its nature, neuronal oscillations can be measured as a mixture of several underlying base frequencies, reflecting certain cognitive, attentional and affective states. These oscillations were defined in specific frequency ranges: Delta, theta, alpha, beta and gamma.
Delta (0.5–4 Hz) is a wave examined during deep-sleep periods, locating its greater-power point in the right hemisphere of the brain. Since sleep is associated with the consolidation of memory, it plays a central role in learning functions. Its amplitude ranges from 20 to 200 μV.Theta (4–8 Hz) is a wave is related to cognitive activities such as selective attention, assimilation of information, processing, learning and working memory. One can have a better record from the prefrontal, central, parietal and temporal zones. Its amplitude ranges from 20 to 100 μV.Alpha is a way related to the oscillatory rhythmic activity when there is a response to sensory stimuli, motor and memory functions. High levels of alpha can be noted when there is relaxation with closed eyes. The suppression of alpha implies a characteristic signature of mental states of interest, e.g., when there is sustained attention to any type of stimulus at a particular time. Its amplitude ranges from 20 to 60 μV and the activation source of these signals is in the occipital region of the brain.Beta (12–25 Hz) is a frequency generated both in the occipital and frontal regions, and it is associated with the active-thinking states, motor functions, visual and spatial coordination, anxiety and concentration. Its amplitude ranges from 2 to 20 μV.Gamma (more than 25 Hz) is a wave tending to have the highest frequency and the lowest amplitude. So far, there is no strong association of what this type of waves reflect in the brain. Some researchers have suggested that the reflection of the interconnection of several sensory responses to an object in a coherent way, therefore, this wave represents an attention process due to intense brain activities. On the other hand, others have thought that it is the effect of neural processes to achieve vision control, and, thereby, it would not represent a cognitive process after all.

Lastly, there is discussion to date about a cerebral rhythm called mu (μ). Its frequency band covers practically the same as alpha, however, it is only observed in the sensory-motor cortex.

There are various devices on the market that can read and process brain signals, each with qualities that differentiate it from its competences, whether due to its cost, number of electrodes, technology used, etc. The main manufacturers of this type of sensor are NeuroSky, Muse, Emotiv, Texas Instrument, among others. See Figure 1 and Table 1. Based on the objectives of this work and analyzing each of the devices available on the market, ThinkGear-AM1 series of NeuroSky company is chosen, despite having only one electrode, it has the necessary development kit for its application and adaptation in a patient or the dolphin, as well as its price since it is cheaper compared to the others. Another distinctive feature is that the parameters necessary for the project can be reconfigured.

One of the drawbacks of ThinkGear-AM1 series of NeuroSky company is that it can only obtain one time series at time, which limits the use of other tools such as brain mapping, since more electrodes are needed to estimate it efficiently.

### 2.2. Electroencephalography Sensor: TGAM1

MindWave Mobile by NeuroSky (Figure 2b) is a Brain Computer Interface (BCI) in a headband that captures the EGG waves (Figure 2a) and the eye blink of an user [10]. This device is useful for developers because they can program powerfull algorithms with an interface connecting to mobile devices easily and being able to use it in research applications [11].

According to Figure 3, NeuroSky Mindwave Mobile can be divided in four parts:Dry Electrode (Figure 3a),Reference and Ground Electrodes (Figure 3b),EEG biosensor TGAM1 (Figure 3c), andCommunication Module (Figure 3c).

NeuroSky’s EEG biosensor digitizes and amplifies raw analog brain signals to deliver concise inputs. The most important features of this biosensor are the following:Direct connection to dry electrode since it does not need a special solution or gel,One EEG channel, Reference, and Ground,Extremely low-level signal detection,Advanced filter with high noise immunity, andRAW EEG at 512 Hz.

The code that may appear in the ThinkGear packets are listed in the Table 2.

TGAM1 has configuration pads to change two default settings applied at chip power up. The configuration pads is located on the backside of the TGAM1, as indicated by the red square in Figure 4a. From Table 3, the BR0 and BR1 pads configure the output baud rate and data content, after the TGAM1 powers up. The M pad configures the notch filter frequency. Normal Output mode includes the following output: Poor quality value, EEG value, Attention value and Meditation value.

A magnified picture of the B1 and B0 pads is shown in Figure 4b. The first row of pads are GND and the third ones are VCC. The TGAM1 output baud rate and data content after power up behavior depends on the pad setting as described in Table 3. For example, the stuff option in the module in Figure 4a has both BR1 and BR0 tie to GND pads for a 9600 baud with Normal Output Mode.

The baud rate can also be configured after the module is powered up by sending commands through the UART interface. The commands are listed in Table 4. When the module is reset, the baud rate setting will revert back to the default set by BR0 and BR1.

TGAM1’s notch filter frequency can be configured with the M configuration pads. It is used to select either 50 Hz or 60 Hz to reduce the AC noise specific to a targeted market. As indicated in Figure 4c, the top pad is GND and bottom pad is VCC. Tie the M pad to VCC pad to select 60 Hz, and to GND pad to select 50 Hz notch filtering frequency.

Unlike the BR0 and BR1 configuration, there is no equivalent software configuration for the M configuration. The most common stuff option for these configuration pads are illustrated in Figure 4a, configuring the TGAM1 for 9600 Baud, normal output and 60 Hz notch filtering frequency.

### 2.3. Time Series Acquisition: DAT-TGAM1 System

The possible brain stimulation by the emitted dolphin-frequency should change the patient’s brain activity, which would be reflected in an EEG. So, we employed a ThinkGear ASIC Module electroencephalographic sensor (TGAM1, Figure 3c), which is an electroencephalographic biosensor responsible for amplifying, processing and digitizing brain activity signals by means of three electrodes—EEG, reference and ground, allowing to collect unprocessed brain wave data.

Besides, we used an algorithm patented by NeuroSky [12] to generate the spectra of the alpha, beta, theta, delta, gamma signals, amplifying the signal of the raw data and eliminating the noise captured during the sampling to apply said algorithm to the remaining signal. On the other hand, the attention and meditation indicators range from zero to 100 depending on how concentrated, stressed, restless, etc. is found the person during the collection of the samples.

The TGAM1 sensor is physically composed of a single channel EGG electrode. Its Bluetooth is used for data transmission, its transmission speed is set to 57,600 bauds. This chip operates within the range 2.97–3.63 V, its maximum input voltage is 10 mV peak to peak, it also has 4 kV electrostatic discharge protection arrangements for contact discharge and 8 kV for air discharge. The transmission speeds at which it can be configured are 1200, 9600 and 57,600 bauds. Its sampling frequency of 512 samples per second, a frequency ranging from 3 to 100 Hz, serial communication, low power consumption 15 mA at 3.3 V. It is also provided with a low-signal quality warning, in addition to having a reduced size and be low-cost [12].

Likewise, the TGMA1 sensor has different connections, but among the most important are the power, transmission, reception and electrode connections. Figure 3c shows the red wire as the main measuring electrode, the green wires are from the ground and reference electrodes, the black wires as the shield of the system electrodes to mitigate as much as possible the ambient noise, the yellow and blue wires are the power and purple is the communication, respectively. The module requires a transmission speed of 9600 bauds and by default it is set to 57,600 bauds because it lacks resistance between pin B1 and pin B0 of the rear part explained below.

The biosignals acquired by TGAM1 are represented in RAW data. The range of Raw Data varies from −32,768 to 32,767, signed 16 bits precision. The first frontal polar electrode, or $F_{p}1$, samples in real-time the discretization of voltage.

This RAW-Volts data relation is defined by Equation (1). Figure 5 shows the representation of the RAW data in μV.
(1)$$Volts=RAWdata\cdot \frac{\frac{1.8}{4096}}{2000}\left[\mu V\right].$$

### 2.4. Filtering Biosignals using Fast Fourier Transform

The Fast Fourier Transform (FFT) algorithm is based on the method known as successive bending. It is defined by Equation (2) as
(2)$$F_{u}=\frac{1}{2\cdot M}\left[\sum_{x=0}^{N-1}\left(a_{x}\cdot w\left(u,x\right)\right)\right]$$
where $w\left(u,x\right)$ can be defined by Equation (3) as
(3)$$w\left(u,x\right)=e^{\left(-j\cdot 2\cdot \frac{u\cdot \pi \cdot x}{2\cdot M}\right)}$$
where the transformation result is represented as a vector $F_{u}$, the value *N* is the sample elements amount, and *M* is half of *N*. The sample values are loaded in the vector $a_{x}$, and *x* represents the subscript. Equations (4) and (5) are obtained from Equation (2) as follows
(4)$$Feven_{u}=\frac{1}{M}\cdot \left[\sum_{x=0}^{M-1}\left(a_{2\cdot x}\cdot w_{1}\left(u,x\right)\right)\right]$$
(5)$$Fodd_{u}=\frac{1}{M}\cdot \left[\sum_{x=0}^{M-1}\left(a_{2\cdot \left(x+1\right)}\cdot w_{1}\left(u,x\right)\right)\right]$$

All the even subscripts of *x* belonging to $a_{x}$ are grouped in the vector Feven and the odd ones in the vector Fodd. In this case, $w_{1}\left(u,x\right)$ is defined by Equation (6) as
(6)$$w_{1}\left(u,x\right)=e^{\left(-j\cdot 2\cdot \frac{u\cdot u\cdot x}{\frac{N}{2}}\right)}$$

Frequently, the relevant information of a signal has a characteristic wave that is known in a general way, so it is possible to generate time series containing this information. Later, one can be able to measure the relations between signals in both the time domain and the frequency (correlation or crossed spectrum) as well as more complex transformations for the characteristics extraction.

In the case of EEG-signal analysis, considering the time plotted on the *X* axis and the voltage on the *Y* axis, FFT transforms the signal from the time domain to the frequency. Basically, what is done during this transformation is to examine how much the raw-data of signal can be approximated to sine waves consisting of pure frequencies, i.e., redundancy-free; the more they adjust, the greater their correlation. Since the frequencies of the human brain associated with affective and cognitive activities range from 0.5 to 60 Hz, an FFT analysis gives a lot of relevant information, e.g., if a person is in a concentration general state (theta band) or the response of its neuronal activity describes a dream state (delta band).

After using Equation (6), a Notch Filter (NF) is employed, which is a Band-stop Filter. NF rejects a band or a specific frequency that could interfere with the EEG sensor, ThinkGear-AM1 series of NeuroSky company. The experiment depicted in Section 3 and Section 4 was realized in Ixtapa, Mexico. Thus, interference frequency of the power line is equal to 60 Hz and the Notch frequency is equal to 60/(512/2) for a bandwidth of 3 dB.

To implement the FFT analysis in the microcontroller it was necessary to apply1 Feven and Fodd. Hence, we used MatLab 2019b to implement both the embedded-devices programming and the analysis of these signals. The treatment of biosignals like EEG requires not only the application of filters to remove noise but also to ensure that the time series of the different patients can be compared each other. That is the reason to normalize the EEG Power Density estimation by Equation (Equation 7), as follows:(7)$$\bar{EEG(\omega )}=\frac{EEG{\left(\omega \right)}_{i}-EEG{\left(\omega \right)}_{min}}{EEG{\left(\omega \right)}_{max}-EEG{\left(\omega \right)}_{min}}$$

Thus, the normalized EEG or $\bar{EEG(\omega )}$ ensures the one dimensional comparison of the samples, whose range will fluctuate between 0 and 1.

### 2.5. Self-Affine Analysis

A self-affine fractal is a set remaining invariant under an anisotropic scale transformation. Many real complex systems can be studied by means of self-affine fractals in a way of long time series recorded.

Many self-affine time series yielded by complex systems display fluctuations on a wide range of time scales following a scaling relation over several or many time scales by fractal scaling exponents. Hence, fractal or Self-Affine Analysis (SSA) has been applied to many biological time series for understanding their dynamics at different time scales [13,14].

The most self-affine time series studied are one dimensional, where the time axis t and the axis of the measured values x(t) are not equivalent; therefore, a rescaling of t by a factor b implies a rescaling of x(t) by a different factor $b^{H}$ to yield the following self-affine equivalence [15]
(8)$$x\left(t\right)=b^{H}x\left(bt\right),$$
where the fractal scaling exponent of Hurst, *H*, points out what kind of self-affinity or correlations are displayed.

The values of *H* are limited to 0<H<1. Values of H<0.5 indicate antipersistent data behavior or negative correlations among them. Values of H>0.5 point out persistent data behavior or positive correlations among them. Moreover, values of H=0.5 show random walk behavior of the data or zero correlations among them.

One way of estimating whether a DAT is efficient or not is to determine if there is a change in the brain activity of patients undergoing a DAT, i.e., to know how the voltage values recorded in the EEG change or fluctuate over time. Hence, the self-affine time series to perform the SSA were those of the voltage fluctuations. Standard deviation is the most used parameter to calculate fluctuation of a variable (in our case, voltage) over time.

The goal of carrying out SSA is to calculate the average value of *H* over many realizations of the time window of different sizes, in order to determine what type of correlations are displayed by the time series fluctuations from the patients underwent a DAT.

In this work, to carried out the SSA, we calculated the structure function of the time series fluctuations. Balankin in [16] pointed out that the long-term memory in the time series fluctuations, F(t+τ), could be studied by means of their structure function, σ(τ,δ*t*),
(9)$$\sigma \left(\tau ,\delta_{t}\right)=\bar{{\langle {\left[F\left(t+\delta_{t},\tau \right)-F\left(t,\tau \right)\right]}^{2}\rangle }^{1/2}}$$
where the overbar denotes average over all *t* in time series of length T−τ (*T* is the length of original EEG time series x(t)) and the brackets denote average over different realizations of the time window of size δ*t*.

Moreover, Balankin in [16] mentioned that the structure function of the time series fluctuations could exhibit the following power law behavior
(10)$$\sigma \propto {\left(\delta_{t}\right)}^{H}$$
where *H* is the Hurst exponent.

## 3. Experiment 1

### 3.1. Initial Conditions

This experiment consisted in testing our BCI by means of capturing the EEG signals of three same age children with Infantile Spastic Cerebral Palsy (ISCP) when they underwent to a Dolphin Assisted Therapy (DAT). The whole system is subdivided into three fundamental parts:Female dolphin of the bottle nose species.Patient with ISCP.DAT-TGAM1 system.

All three subsystems interact to determine whether or not a DAT is effective for ISCP patients. For analyzing EEG data, we decomposed the EEG signal into functionally distinct frequency bands: δ (0.5–4 Hz), θ (4–8 Hz), α (8–12 Hz), β (12–30 Hz), γ (30–60 Hz), and All Bands (0.5–60 Hz) by applying FFT, in order to obtain an estimate of the Power Spectral Density (PSD), expressed in μV/Hz [17]. Later, we explored the behavior of the three ISCP patients using FFT and SAA from EEG signals [18]. The EEG RAW data were time series showing the cerebral brain activity (voltage versus time) At rest or before (Figure 6a) and during DAT (Figure 6b) for ISCP patients, as recorded by the first frontopolar electrode ($F_{P1}$) by means of a EEG biosensor TGAM1 Module, Figure 3c [19].

Written informed consent was obtained from every individual prior to taking part in the study, according to CONFIDENTIALITY COMMITMENT LETTER of Instituto Politécnico Nacional and the institutional ethics committee. Every participant was informed about the possibility of withdrawing from the experiment without any disadvantage, signed a written informed consent form accordingly (30 October 2018). The Ethics Committee of the National Polytechnic Institute of Mexico approved the methodology for sampling and treating female bottlenose dolphins (Tursiops truncatus) in document number D/1477/2020.

### 3.2. Results

Figure 7 shows the results after carrying out Experiment 1 to measure the DAT efficiency for the RAW EEG Brain activity (first row) of three child patients with ISCP (columns) using FFT-PSD (second and third rows) and SSA (last two rows) before and during DAT.

On the one hand, Figure 7a–c show a higher brain activity during DAT than before. Likewise, the FFT analysis points out an average great increment of 298.74% $\left(\frac{5456914.31}{1368533.76}-1\right)$ in the average power spectral density of data during DAT respect to before, causing an increment on every spectral band, Figure 7d–f. Moreover, Figure 7g–i show the entire power spectral density from 0 to 256 Hz (half of sampling frequency $\delta_{t}/2$), where it can be noticed that the overall power is higher in all cases when a DAT was performed.

On the other hand, to carry out the SSA we made the following. From each original EEG time series x(t), we constructed 198 time series of standard deviations (or fluctuations), F(t+τ), for before and other 198 times series F(t+τ) for during DAT. Therefore, we constructed a total of 396 time series F(t+τ) for each one patient with ISCP (Figure 7a, for Patient 1 with ISCP, Figure 7b, for Patient 2 with ISCP, and Figure 7c, for Patient 3 with ISCP)). In Figure 7m, we show the following crossovers: $\tau_{B}=105$ and $\tau_{D}=145$; for Figure 7n, $\tau_{B}=78$ and $\tau_{D}=141$; and for Figure 7o, $\tau_{B}=88$ and $\tau_{D}=114$. A crossover is the point in time where the system whole behavior changes; in our case, it was the τ where the time series F(t+τ) left to follow a power law (or fractal) behavior.

To perform the fractal analysis of time series F(t+τ), we took into account the length of each of the before time series $x_{B}\left(t\right)$ for one minute, $T_{B}$ = 27,678 voltage records (μV) versus time (s), and the length of each of the during DAT time series $x_{D}\left(t\right)$ for five minutes, $T_{D}$ = 151,170 voltage records versus time, with Δt=1/512 s sampling frequency. Moreover, we considered a range of the time interval of the sample of 3≤τ≤200 for the 396 time series F(t+τ), with time windows of the intervals samples $\delta_{t}=1/512$ s.

Based on the above, we estimated the structure function $\sigma \left(\tau ,\delta_{t}\right)=\bar{{\langle {\left[F\left(t+\delta_{t},\tau \right)-F\left(t,\tau \right)\right]}^{2}\rangle }^{1/2}}$ versus $\delta_{t}$ for 3≤τ≤200, for the three patients with ISCP, before and during DAT (Figure 7j–l, and Table 5).

Respect to the first patient with ISCP, for before and during DAT, the average values of the exponent *H* were ${\bar{H}}_{B}=0.4652$ and ${\bar{H}}_{D}=0.3883$, respectively, displaying in all cases negative correlations in the short-term or antipersistent behavior, Figure 7j. Regarding the second patient with ISCP, for before and during DAT, the average values of the exponent *H* were ${\bar{H}}_{B}=0.3279$ and ${\bar{H}}_{D}=0.4685$, respectively, displaying in all cases also antipersistent behavior, Figure 7k. Lastly, respect to the third patient with ISCP, for before and during DAT, the average values of the exponent *H* were ${\bar{H}}_{B}=0.2312$ and ${\bar{H}}_{D}=0.3546$, respectively, displaying in all cases also negative correlations in the short-term or antipersistent behavior, Figure 7l.

### 3.3. Discussion

On the one hand, we analyzed the results individually for each patient with ISCP. It is important to mention that all the patients are intervention subject and Patients 1 and 3 are males, while Patient 2 is female, all in the same age range. Figure 7j–i indicate that the δ band predominates in all patients and reports a considerable increase of more than 100% $\left(\frac{2\times 10^{7}}{1\times 10^{7}}\right)$, which leads to longer learning periods. For Patient 1, there was an increment in power density of 219.58% $\left(\frac{3796590.98}{1188001.28}-1\right)$. While for Patient 2 there was an increment very similar to Patient 1 of 215.86% $\left(\frac{4399351.94}{1392800.00}-1\right)$. All patients experienced an increment in brain activity greater than 200%, but Patient 3 reported an increment of 436.12% $\left(\frac{8174800.00}{1524800.00}-1\right)$, showing a modification in neuronal behavior of all patients by an external factor when they were in dolphin-assisted therapy.

On the other hand, after applying the SSA to the time series F(t+τ) of each patient with ISCP for estimating their structure function (Equation (9)), the results point out that in the three patients with ISCP an antipersistent behavior is exhibited and characterized by $\sigma \propto {\left(\delta_{t}\right)}^{H}$, for both before and during DAT (Figure 7j–l and Table 5). This means that if the brain activity is rising, it was more probable the next voltage value was lower, and vice versa.

For Patient 1 with ISCP, the value of the Hurst exponent, *H*, was reduced 16.53% from before (${\bar{H}}_{B}=0.4652$) to during DAT (${\bar{H}}_{D}=0.3883$). Hence, the grade of antipersitent in this patient was raised, i.e., the probability that was increased that the next voltage value was lower by having an upward trend in the patient’s brain activity, and vice versa.

For Patient 2 with ISCP, the value of *H* was increased 42.88% from before (${\bar{H}}_{B}=0.3279$) to during DAT (${\bar{H}}_{D}=0.4685$). Therefore, the grade of antipersitent in this patient was reduced, i.e., the probability was reduced that the next voltage value was lower by having an upward trend in the patient’s brain activity, and vice versa.

For Patient 3 with ISCP, the value of *H* was increased 53.37% from before (${\bar{H}}_{B}=0.2312$) to during DAT (${\bar{H}}_{D}=0.3546$). Thereby, the grade of antipersitent in this patient was reduced, i.e., the probability was reduced that the next voltage value was lower by having an upward trend in the patient’s brain activity, and vice versa.

## 4. Experiment 2

### 4.1. Initial Conditions

For this experiment, we explored the behavior of the control and intervention patients by using the FFT and SSA mathematical tools applied to their EEG signals [18]. The EEG RAW data were time series showing their cerebral brain activity (voltage versus time) At rest or before DAT (Figure 10a), during DAT (Figure 10b), and after DAT (Figure 8c [19]). All EEG were recorded from the first frontopolar electrode ($F_{P1}$) by means of an EEG biosensor TGAM1 Module.

This experiment consisted in testing our BCI by means of capturing the EEG signals of two same age children, one of them with Obsessive Compulsive Disorder (OCD) and the other one as the control patient, when they underwent to a DAT. The whole system is subdivided into four fundamental parts:Female dolphin of the bottle nose species.Control Patient.Invervention Patient with OCD.DAT-TGAM1 system.

All four subsystems interact to determine whether or not a DAT is effective for OCD patients regarding neurotypical or control patient. Written informed consent was obtained from every individual prior to taking part in the study, according to CONFIDENTIALITY COMMITMENT LETTER of Instituto Politécnico Nacional and the institutional ethics committee. Every participant was informed about the possibility of withdrawing from the experiment without any disadvantage, signed a written informed consent form accordingly (17 May 2019). The Ethics Committee of the National Polytechnic Institute of Mexico approved the methodology for sampling and treating female bottlenose dolphins (Tursiops truncatus) in document number D/1477/2020.

### 4.2. Results

Figure 9 shows the results after carrying out Experiment 2 to measure the DAT efficiency for RAW EEG Brain activity (first row) of the control and intervention child patients, first and second columns respectively, using FFT-PSD (second and third rows) and SSA (last two rows) before DAT, during DAT and after DAT.

On the one hand, we analyzed the results individually for each patient. It is important to mention that Patient 2 was an intervention subjec and is male, while Patient 2 is female and can be considered as a control subject, all in the same age range. For Patient 1, Figure 9c shows that the fast frequencies, that is, the β and γ bands, predominated with respect to the slow frequencies. Hence, the processes of active-cognitive thought, coordination and concentration were increased by 376.28% $\left(\frac{78058500}{16389000}-1\right)$ during DAT. From Figure 9d, Patient 2 reacted in the opposite way, since she presented a higher excitation in the slow frequencies, i.e., the δ and θ bands. However, if we compare their increment of the average power density during DAT with respect to the initial state of the patient, we notice that the low frequencies increased 3.33 times $\left(\frac{0.95\times 10^{8}}{0.285\times 10^{8}}\right)$ while the fast frequencies had increased 5.45 times $\left(\frac{0.9\times 10^{8}}{0.165\times 10^{8}}\right)$. According to Fitzgerald and Brendon in [20], an increase in power density in the β and γ bands leads to a balance between inhibition and excitation. Thus, Patient 2 increased her power density in both the β and γ bands significantly, leading us to indicate that the therapeutic effects necessary regulate her brain activity occurring during DAT.

On the other hand, to carry out the SSA we made the following. From each original EEG time series x(t), we constructed 198 time series of standard deviations (or fluctuations), F(t+τ), for before, other 198 times series F(t+τ) for during DAT and other other 198 times series F(t+τ) for after. Therefore, we constructed a total of 594 time series F(t+τ) for each patient (Figure 9a, for control patient or Patient 1, and Figure 9b, for patient with OCD or Patient 2). In Figure 9i we show the following crossovers for the control patient: $\tau_{B}=33$, $\tau_{D}=127$ and $\tau_{A}=188$; and in Figure 9j for the patient with OCD: $\tau_{B}=119$, $\tau_{D}=99$ and $\tau_{A}=136$.

To perform the fractal dynamics of the time series F(t+τ), we took into account the length of each of the before time series $x_{B}\left(t\right)$ during one minute, $T_{B}$ = 27,678 voltage records (μV) versus time (s), the length of each of the during DAT time series $x_{D}\left(t\right)$ during five minutes, $T_{D}$ = 151,170 voltage records versus time, and the length of each of the after time series $x_{A}\left(t\right)$ during one minute, $T_{A}$ = 27,678 voltage records versus time, with Δt=1/512 s sampling rate. Moreover, we considered a range of the time interval of the sample of 3≤τ≤200 for the 594 time series F(t+τ), with time windows of the intervals samples $\delta_{t}=1/512$ s.

Based on the above, we estimated the structure function $\sigma \propto {\left(\delta_{t}\right)}^{H}$ versus $\delta_{t}$ for the control patient and the patient with OCD, 3≤τ≤200, before, during DAT and after (Figure 9g,h and Table 6).

Respect to the control patient, for before, during DAT and after, the average values of the exponent *H* were ${\bar{H}}_{B}=0.1978$, ${\bar{H}}_{D}=0.2178$ and ${\bar{H}}_{A}=0.2151$, respectively, displaying in all cases negative correlations in the short-term or antipersistent behavior, Figure 9g. Regarding the patient with OCD, for before, during DAT and after, the average values of the exponent *H* were ${\bar{H}}_{B}=0.4405$, ${\bar{H}}_{D}=0.4572$ and ${\bar{H}}_{A}=0.4235$, respectively, displaying in all cases also antipersistent behavior, Figure 9h.

### 4.3. Discussion

Table 6 summarizes our experimental results. When we compared Average Power Spectrum Density (PSD) during DAT regarding before, we can realize that the patients increment their brain activity in approximately 376.28% $\left(\frac{78058500}{16389000}-1\right)$ in both cases. Comparing after respect to during DAT, we found that the brain activity of the control patient was reduced in 67.92% $\left(\frac{23470000}{73158000}-1\right)$, while the intervention patient reduced his brain activity in 83.02% $\left(\frac{14088000}{82959000}-1\right)$, namely, the patient with OCD got an apparent relaxation since its PSD decreased 19.19% $\left(\frac{14088000}{17433000}-1\right)$ regarding its initial rest state meanwhile the control patient increased her PSD 52.95% $\left(\frac{23470000}{15345000}-1\right)$ respect to her initial rest state.

On the one hand, Figure 9a,b show a higher brain activity during DAT than before or after. Evenly, the FFT analysis points out an average great increment of 376.28% $\left(\frac{78058500}{16389000}-1\right)$ in the average power spectral density of data in both patients during DAT respect to before DAT, yielding an increment on every spectral band, Figure 9c,d. Likewise, Figure 9e,f show the entire power spectral density from 0 to 256 Hz (half of $\delta_{t}$), where it can be noticed that the overall power was higher in all cases when a DAT was carried out.

On the other hand, after applying the SSA to the time series F(t+τ) of the control and intervention (OCD) patients for estimating their structure function (Equation (9)), the results point out that in the two patients also an antipersistent behavior was exhibited and characterized by $\sigma \propto {\left(\delta_{t}\right)}^{H}$, for before, during DAT and after (Figure 9g,h and Table 6). This means that if the brain activity was rising, it was more probable the next voltage value was lower, and vice versa.

For the control patient or Patient 1, the value of the Hurst exponent, *H*, was increased 10.11% from before (${\bar{H}}_{B}=0.1978$) to during DAT (${\bar{H}}_{D}=0.2178$); thus, the grade of antipersitent in this patient was reduced, i.e., the probability was decreased that the next voltage value was lower by having an upward trend in the patient’s brain activity, and vice versa. In addition, the value of *H* was reduced 1.24% from during DAT (${\bar{H}}_{D}=0.2178$ to after (${\bar{H}}_{A}=0.2151$); thus, the grade of antipersitent in this patient was increased a little, i.e., the probability was increased a little that the next voltage value was lower by having an upward trend in the patient’s brain activity, and vice versa.

For the patient with OCD or Patient 2, the value of *H* was increased 3.79% from before (${\bar{H}}_{B}=0.4405$) to during DAT (${\bar{H}}_{D}=0.4572$); thus, the grade of antipersitent in this patient also was reduced. Moreover, the value of *H* was reduced 0.81% from during DAT (${\bar{H}}_{D}=0.4572$ to after (${\bar{H}}_{A}=0.4235$); therefore, the grade of antipersitent in this patient was also increased a little. Table 6 shows these results.

## 5. General Discussion

Firstly and based on the outcomes from the PSD analysis, we obtained the results of Table 6, showing a huge increment of brain activity during DAT regarding the other two states, before and after therapy. These experiments show the efficiency of DAT was around 376.28% $\left(\frac{78058500}{16389000}-1\right)$ of increment of neuronal PSD. The novelty of the PSD analysis in a TGAM1 EEG sensor is the ability to identify and give a mathematical meaning of the beneficial changes in behavior of the neuronal system of children with or without Obsessive Compulsive Disorder (OCD) when their treatment is assisted by a bottle-nose dolphin. It is important to highlight that the increment of neural activity (more PSD) during DAT, displaying collective behaviors, i.e., positive increments of neuronal activity could be followed by much more neural activity. Thus, the brain could react to DAT gradually over a period of time, which indicates the existence of external features that modify the brain behavior of children with diverse psychological and physical disabilities or disorders.

Regards to the SSA, we found the following. In Experiment 1, the value of the exponent *H*, from before to during DAT, was reduced 16.53% for the Patient 1 and increased 42.88% and 53.37% for Patients 2 and 3, respectively. Therefore, DAT was efficient in two of out the three patients with ISPC, because in during DAT the fluctuation of voltage tended to be explained by a power law.

Besides, in Experiment 2, the value of the exponent *H*, from before to during DAT, was increased 10.11% and 3.78% for the control patient and the patient with OCD, respectively. Moreover, the value of *H*, from during DAT to after, was reduced 1.24% and 0.81% for the control patient and the patient with OCD, respectively. While the value of *H* practically did not change for both patients, DAT was more efficient for the patient with OCD because its value of *H* was 0.4572 and for the control patient was 0.2178 during DAT. Hence, DAT helped patient with OCD to keep a relax state.

Based on the outcomes obtained from to the SSA, we can claim that the structure function (Equation (9)) of the three patients with ISCP, the control patient and the patient with OCD displayed an antipersistent behavior, characterized by $\sigma \propto {\left(\delta_{t}\right)}^{H}$, for before, during DAT and after. This means that if the brain activity of all five patients was rising, it is more probable their next voltage value was lower, and vice versa.

In the aftermath of the above, we decided to graph the whole fluctuation time series for both patients in Experiment 2 because we had data of before, during DAT and after, in order to determine where there was a behavior fitted by a power law, see Figure 10a,b. In spite of getting a value of the exponent *H* lower than 0.5 (antipersistent behavior) in both cases, we can observe that almost both curves are fitted to a power law (straight line). If we erase the first five points and the last ten points in the curve from Figure 10a and the last ten points from Figure 10b, i.e., leaving only practically the points corresponding to during DAT, the adjusting to the power law could be improved dramatically, and therefore raising the value of *H* above 0.5, obtaining positive correlations or persistent behavior during DAT.

Another way to assess whether this alternative therapy is being efficient to the patients is to increase the during DAT time when the samples are collected, i.e., extending the time series to be studied. According to Figure 10a,b, we suppose the fitting of data to a power law (straight line) will raise over the time, and hence arising the value of the exponent *H* above 0.5 (persistent behavior), until appear a crossover and the curve tends to be horizontal, pointing out that our system has reached a stationary state where fluctuations will not almost change. Latter the aforementioned crossover it would not worth to continue the DAT.

## 6. Conclusions

Brain Computer Interface (BCI) can have a variety of application areas such as Neuroscience research. For instance, BCI can help Infantile Spastic Cerebral Palsy (ISCP), Obsessive Compulsive Disorder (OCD) and neurotypic children where its sources mainly include EEG signals obtained from brain using head sensors such as TGAM1, that is, ThinkGear-AM1 series of NeuroSky company applied to develop the proposed system in this paper.

In the present work reduces the input noise was reduced by means of a Notch Filter, so for future works it would be recommended to use filters based on Evolutionary Computation Algorithms, like the one presented in [21], which would allow not only to extract characteristics within the EEG but also additive noises.

Based on the Raw EEG records generated by the BCI proposed, the Self-Affine and Power Spectrum analysis were useful mathematical tools to assess the efficiency Dolphin-Assisted Therapy (DAT) by studying the structure function of the brain activity fluctuations from the five child patients at different time-scales, as well as the increase in spectral density of brain bands, considered as biomarkers with a therapeutic effect.

On the one hand, from the outcomes obtained by the PSD analysis (Table 6), we observe in the three child patients with ISCP (Experiment 1) and the child patient with OCD (Experiment 2) a huge increment of brain activity during DAT regarding the other two states, before and after therapy around 376.28% $\left(\frac{78058500}{16389000}-1\right)$. The novelty of the PSD analysis in a TGAM1 EEG sensor is the ability to identify and give a mathematical meaning of the beneficial changes in behavior of the neuronal system of children with or without ISCP and/or OCD when their treatment is assisted by a bottle-nose dolphin. This increment of neural activity (more PSD) during DAT displayed positive increments of neuronal activity followed more probably by much more neural activity. Thus, the child patients’ brain could react to DAT gradually over a period of time. Hence, one challenge is to determine at what point of time the therapy should be suspended.

On the other, the results yielded by the Self-Affine analysis (Table 6) point out that the structure function (Equation (9)) of the three patients with ISCP (Experiment 1), the control patient and the patient with OCD (Experiment 2) displayed an antipersistent behavior, characterized by $\sigma \propto {\left(\delta_{t}\right)}^{H}$, for before, during DAT and after. This means that if the brain activity of all five patients was rising, it is more probable their next voltage value was lower, and vice versa.

We graphed the whole fluctuation time series for both patients in Experiment 2 because we had data of before, during DAT and after. In spite of getting a value of the exponent *H* lower than 0.5 (antipersistent behavior) in both cases, we observe that almost both curves are fitted to a power law (straight line) (Figure 10a,b). If only we leave practically the points corresponding to during DAT, it could be improved dramatically the adjusting to power law, and therefore raising the value of *H* above 0.5, obtaining positive correlations or persistent behavior during DAT.

Another way to assess whether this alternative therapy is being efficient to the child patients is to increase the during DAT time when the samples are collected, supposing the fitting of data to a power law (straight line) will raise over the time, and hence arising the value of the exponent *H* above 0.5 (persistent behavior), until appear a crossover and the curve tends to be horizontal, pointing out that our system has reached a stationary state where fluctuations will not almost change. Latter the aforementioned crossover it would not worth to continue the DAT.

Nowadays is impossible to collect samples extending the time for during DAT due to the Covid-19 pandemic. Therefore, it could be possible to generate artificial biosaignals or self-affine traces based on raw EEG recordings for the during DAT with the *H* values above 0.5 as follows.

First, the average voltages are generated as follows. Obtain the average of voltage 1 and voltage 2, then the average from voltage 1 to voltage 3, then the average from voltage 1 to voltage 4, and so on to the average from voltage 1 to the last voltage. Thus, n−1 voltage averages are calculated.

Secondly, the value of the Hurst exponent *H* that is desired to have the structure function of voltage fluctuations is established, which will always be greater than 0.5 and less than one.

Third, the range of voltage fluctuations are determined for every 512 data (one second), that is, the maximum fluctuations that could be had in this time interval. This is done by calculating the standard deviation for every 512 data, i.e., for the first point it is considered from voltage 1 to voltage 512, for point two from voltage 2 to voltage 513, for point three from voltage 3 to voltage 514, and so on until the last point from n−512 voltage to last voltage. In this way n−512 data of standard deviations or fluctuations are obtained. Next, the maximum and minimum values of these n−512 data are calculated, and then the range of these data is estimated (maximum value minus minimum value).

Finally, to generate the probabilistic scenarios for during DAT, *m* self-affine time series (or curves) will be created randomly based on the Brownian motion theory and from the three parameters previously estimated: Voltage averages, value of the exponent *H* and range of fluctuations. Each of the generated self-affine *m* time series would consist of n−1 voltage data. Each data indicates the predicted average values of the voltage fluctuation, during an interval of one second (512 voltage-records) and with the previously defined range.

## Figures and Tables

**Figure 1 brainsci-10-00403-f001:**
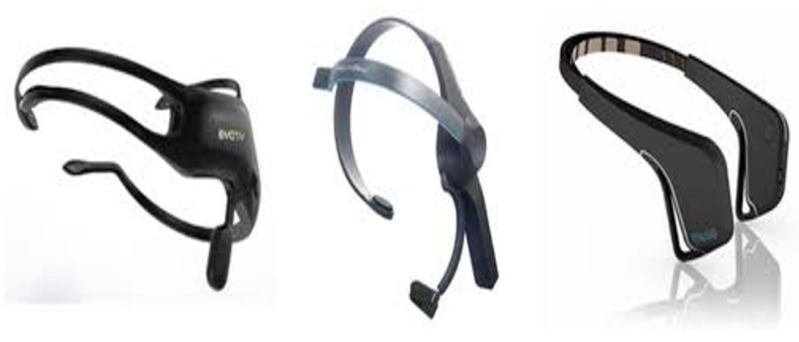
Electroencephalographic devices.

**Figure 2 brainsci-10-00403-f002:**
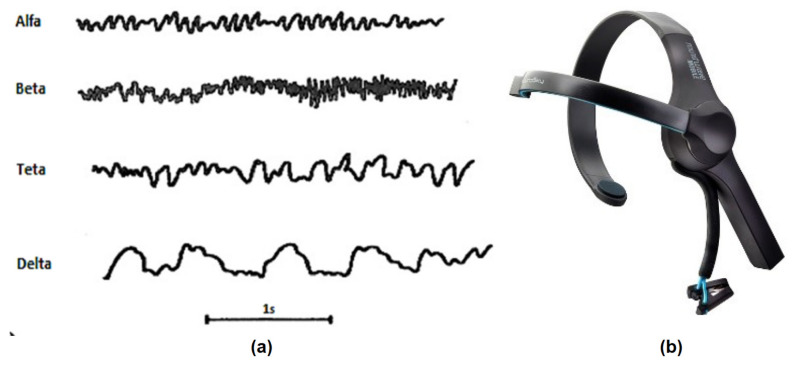
(**a**) Types of Electroencephalogram (EEG) waves and (**b**) NeuroSky Mindwave Mobile.

**Figure 3 brainsci-10-00403-f003:**
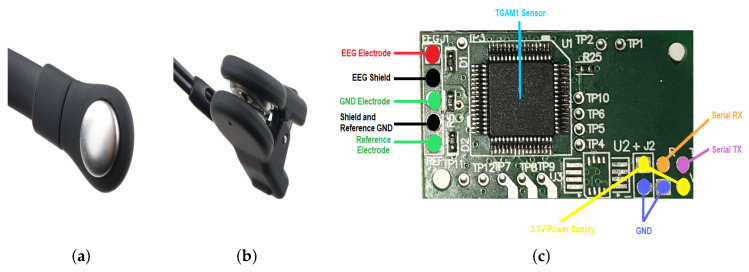
NeuroSky Mindwave Mobile system main parts. (**a**) Dry Electrode. (**b**) Reference and Ground Electrodes. (**c**) EEG biosensor TGAM1 with Communication Module.

**Figure 4 brainsci-10-00403-f004:**
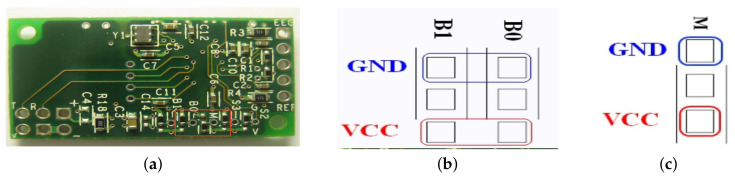
Configuration pads in TGAM1 chip. (**a**) Main pads. (**b**) B0 and B1 pads. (**c**) M pad.

**Figure 5 brainsci-10-00403-f005:**
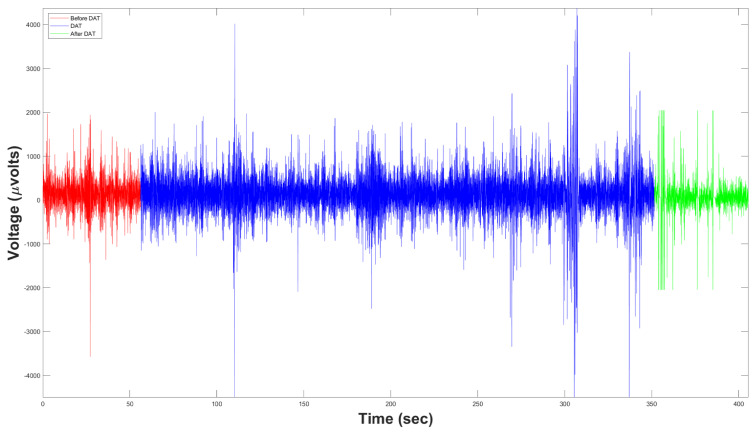
Representation of the RAW data in μV.

**Figure 6 brainsci-10-00403-f006:**
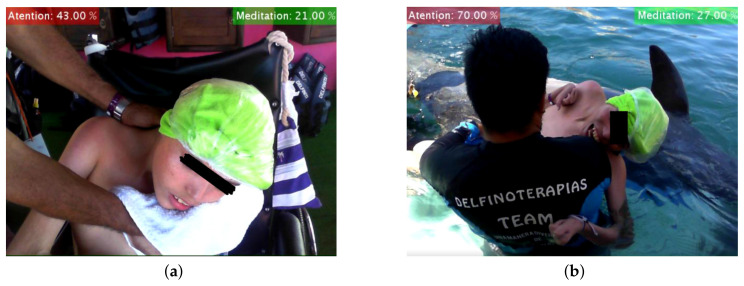
Method for obtaining EEG RAW samples. (**a**) ISCP patient At Rest or Before. (**b**) ISCP patient During a Dolphin-Assisted Therapy.

**Figure 7 brainsci-10-00403-f007:**
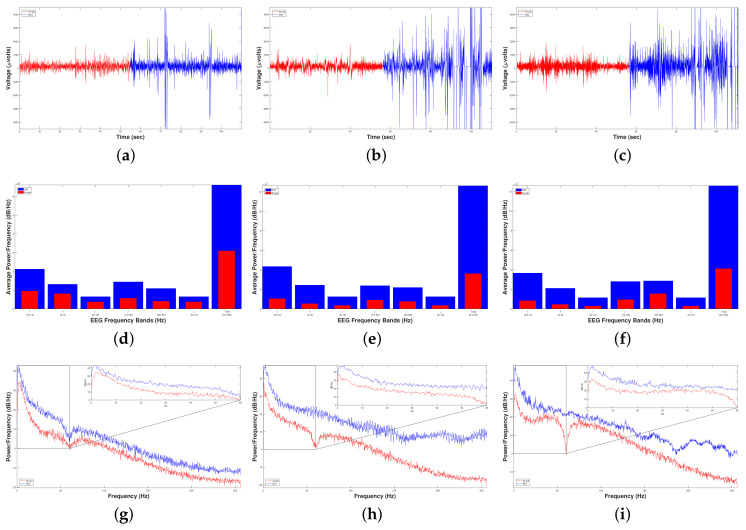
Results of Dolphin-Assisted Therapy (DAT) efficiency for RAW EEG Brain activity (first row, (**a**–**c**)) of three child patients with Infantile Spastic Cerebral Palsy (ISCP) (each column is a patient). Periodogram (second row, (**d**–**f**)) and Average Spectogram (third row, (**g**–**i**)) using FFT Power Spectrum Density. Fluctuations EEG Brain activity (fourth row, (**j**–**l**)) and Crossover (fifth row, (**m**–**o**)) using Self-Affine Analysis. The functions in red mean that the sample was taken before DAT, while the blue ones mean during DAT.

**Figure 8 brainsci-10-00403-f008:**
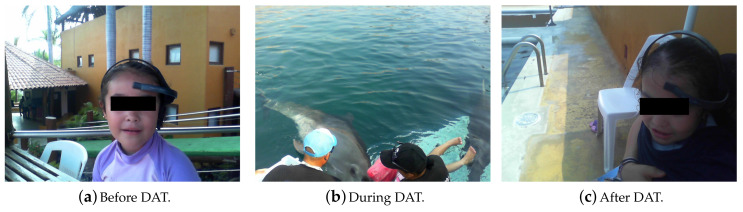
Method of obtaining EEG RAW samples.

**Figure 9 brainsci-10-00403-f009:**
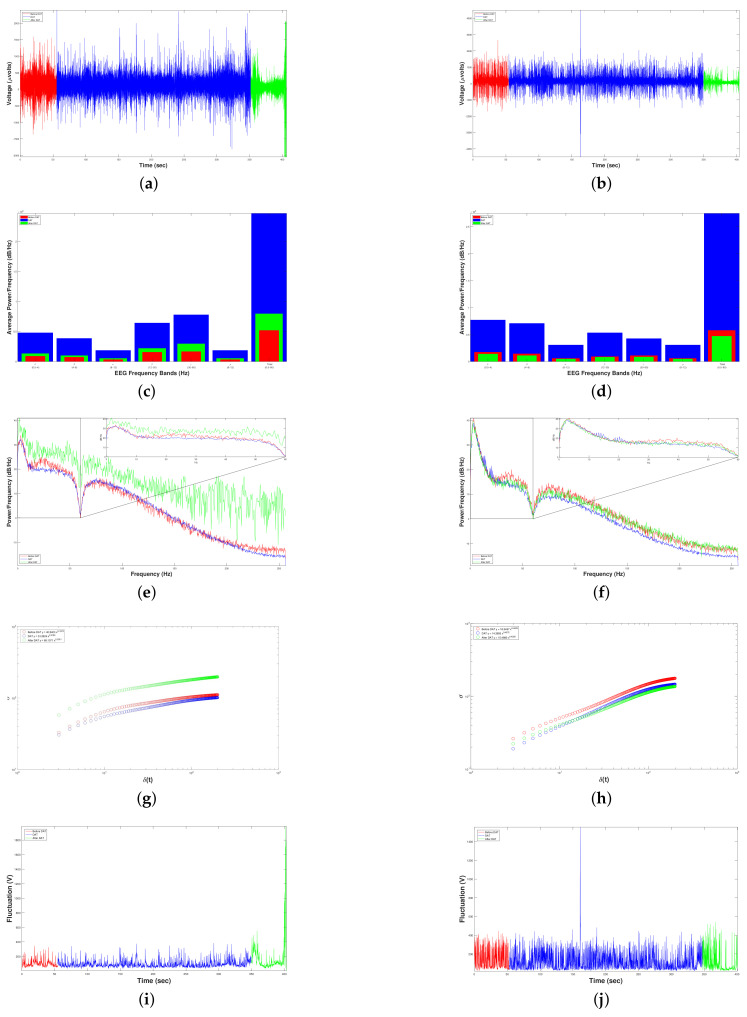
Results of DAT efficiency for RAW EEG Brain activity (first row, (**a**,**b**)) of two child patients (each column is a patient). Periodogram (second row, (**c**,**d**)) and Average Spectogram (third row, (**e**,**f**)) using FFT Power Spectrum Density. Fluctuations EEG Brain activity (fourth row, (**g**,**h**)) and Crossover (fifth row, (**i**,**j**)) using Self-Affine Analysis. The functions in red mean the sample taken before DAT, while blue and green ones mean during DAT and after DAT, respectively.

**Figure 10 brainsci-10-00403-f010:**
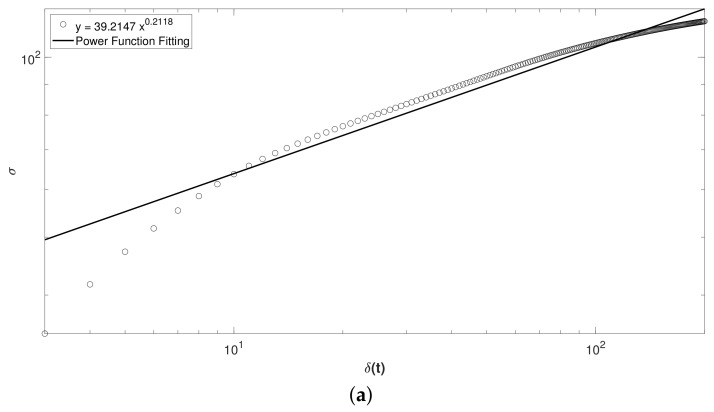
Whole Self-affine analysis of: (**a**) Control patient and (**b**) Intervention Patient with Obsessive Compulsive Disorder.

**Table 1 brainsci-10-00403-t001:** Comparison of electroencephalographic biosensors.

Model	Muse	EOTIV Insight	EMOTIV EPOC+	TGAM1
**Brand**	InteraXon	Emotiv	Emotiv	NeuroSky
**Release**	abr-14	ago-15	dic-09	mar-11
**Sensors**	EEG	EEG	EEG	EEG
**Electrodes**	14	5	14	1
**ADC**	16 bits	14 bits	16 bits	12 bits
**Rank**	N/A	1–43 Hz	0.2–43 Hz	3–100 Hz
**Frequency**				
**Frequency of Sampling**	220–500 Hz	128 Hz	128–256 Hz	512 Hz
**Voltage Entry**	2 mV	0.51 V	8.4 mV	1 mV
**Autonomy**	5 Hrs	4 Hrs	12 Hrs	8 Hrs
**SDK**	Si	Si	Si	Si
**Platform**	Android iOS	MacOS Windows Linux	MacOS Windows Linux	MacOS
		Android	Android	Windows Android
		iOS	iOS	iOS
**Interface**	Bluetooth 2.1	Bluetooth 4.0	Bluetooth 4.0	Bluetooth 2.1
**Price**	249 USD	299 USD	799 USD	50 USD

**Table 2 brainsci-10-00403-t002:** ThinkGear CODE.

Code	Length	Value	Default Setting
0x02	N/A	Poor Quality (0–200)	On
0x04	N/A	eSense Attention (0–100)	On
0x05	N/A	eSense Meditation (0–100)	On
0x80	2	10-bit Raw EEG	Off
0x83	24	EEG Powers (integer)	On

**Table 3 brainsci-10-00403-t003:** Configuration pads.

BR1	BR0	Function
GND	GND	9600 Baud with Normal Output Mode
GND	VCC	1200 Baud with Normal Output Mode
VCC	GND	57.6k Baud with Normal + Raw Output Mode
VCC	VCC	N/A

**Table 4 brainsci-10-00403-t004:** Baud rate configuration.

Command	Function
0x00	9600 Baud with Normal Output Mode
0x01	1200 Baud with Normal Output Mode
0x02	57.6k Baud with Normal + Raw Output Mode

**Table 5 brainsci-10-00403-t005:** Summary of the experimental results of Experiment 1.

Patient	Power Spectrum Density		Self-Affine Analysis			
	BEFORE	DAT	BEFORE		DAT	
			$\bar{H}$	CrossOver	$\bar{H}$	CrossOver
1	1188001.28	3796590.98	0.4652	105.00	0.3883	145.00
2	1392800.00	4399351.94	0.3279	78.00	0.4685	141.00
3	1524800.00	8174800.00	0.2312	88.00	0.3546	114.00
**Average**	**1368533.76**	**5456914.31**	**0.3414**	**90.33**	**0.4038**	**133.33**

The data in red means that before DAT, while the blue ones mean during DAT.

**Table 6 brainsci-10-00403-t006:** Summary of the experimental results of Experiment 2.

Patient	Power Spectrum Density			Self-Affine Analysis					
	BEFORE	DAT	AFTER	BEFORE		DAT		AFTER	
				$\bar{H}$	CrossOver	$\bar{H}$	CrossOver	$\bar{H}$	CrossOver
1	15345000	73158000	23470000	0.1978	33.00	0.2178	127.00	0.2151	185.00
2	17433000	82959000	14088000	0.4405	119.00	0.4572	99.00	0.4235	136.00
**Average**	**16389000**	**78058500**	**18779000**	**0.3192**	**76.00**	**0.3375**	**113.00**	**0.3193**	**160.50**

The data in red means that before DAT, the blue ones mean during DAT, and the green ones mean after DAT.

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
