# Peer review of "Neurodynamics of Patients during a Dolphin-Assisted Therapy by Means of a Fractal Intraneural Analysis"

_brainsci, 2020, doi:10.3390/brainsci10060403_

Round 1

Reviewer 1 Report

Brain Sciences

Sections:

Theoretical & Computational Neuroscience

Behavioral and Cognitive Neurodynamics

Neurodynamics of patients during a dolphin-assisted therapy by means of a fractal intraneural analysis

Oswaldo Morales Matamoros, Jesús Jaime Moreno Escobar*, Ricardo Tejeida Padilla, Ixchel Lina Reyes

COMMENTS/SUGGESTIONS FOR AUTHORS

MAIN OUTCOMES REPORTED IN THIS MANUSCRIPT

In this manuscript, the authors provide a biosensor technology application (so-called EEG biosensor TGAM1) designed specifically to control brain-computer interaction during Dolphin-Assisted Therapy (DAT) in patients with Infantile Spastic Cerebral Palsy (ISCP, Experiment 1), as well as, with Obsessive Compulsive Disorder (OCD, Experiment 2). This study includes the analysis and interpretation of the acquired EEG biosignals before, during and/or after a DAT. The DAT-TGAM1/EEG system is a good example of technology application with repercussions in the treatment of neuropathologies. According to the authors, DAT improves and reinforces other therapies without replacing them, and in my opinion, it is a striking way of non-invasive neuromodulation that could serve as an alternative to other neuromodulation techniques (e.g., tDCS, TMS). Furthermore, the authors make a detailed examination on brain disorders and diseases that can actually be treated with DAT and make proper exposure of the advantages of studying brain waves as a marker of cognitive changes during and/or after DAT. Anyhow, I have some concerns about the proposed brain-computer system (DAT-TGAM1/EEG) based on biosignals and on the neurodynamic approach by fractal analysis. In my opinion, this manuscript would deserve publication in Brain Sciences once the following concerns/questions are addressed.

BCI (DAT-TGAM1/EEG SYSTEM) AND BIOSIGNAL PROCESSING

[P1] From an analytical and methodological point of view, the two new contributions of this study, compared to another (Moreno J et al. Design of a Brain-Computer System to Measure Brain Activity during a Dolphin-Assisted Therapy using the TGAM1 EEG Sensor. Research in Computing Science 148(11): 139-151, 2019) were the extension to other neuropathology (ISCP: Infantile Spastic Cerebral Palsy) and the application of a neurodynamic approach using self-affine fractal analysis (SSA) for both OCD (Obsessive Compulsive Disorder) and ISCP. Please, include the aferomentioned paper from your group in the Reference section of this manuscript.

[P2] The goal of carrying out SSA was to calculate the average value of H (Hurst exponent) over many realizations of the time window of different sizes, in order to determine what type of correlations (negative, H < 0.5, antipersistent data behavior; OR positive, H > 0.5, persistent data behavior) were displayed by the time series fluctuations from the patients underwent a DAT. In Figs. 6 and 8 and Tables 4 and 5, the results point out that in each patient (3 patients with ISCP and 1 patient with OCD) an antipersistent behavior (H < 0.5 and negative correlation) was exhibited and characterized by the power law among the structure-function fluctuation, the time-window size, and the exponent H (Balakin, 2007), for the different phases: before, during and/or after DAT in the two experiments. The main problem here is that the comparisons were intra-patient and inter-phases (before, during and/or after).

[Q1] Why a preliminary system calibration was not performed to establish very similar (in statistical sense) H-values in the "Before"-phase of DAT for all patients in each of the experiments? This calibration control of the BCI system (DAT-TGAM1/EEG), in parallel with a Z-score analysis for both PSD (by FFT) and H (by SSA) values from the EEG recordings of the patients would allow determining the actual performance of the DAT with relevance intra–patient and inter–patients.

[Q2] In the Conclusion section, the author claim that echolocation is considered as a feature or external element that helps have therapeutic effects in patients in the long-term (lines 502-504, page 16). Which specific result of those shown in this manuscript allows the authors to reach this conclusion? I do not see clear, please explain in more detail (by mean of Figures or Tables) each statement that is made.

[Q3] In General Discusion and Conclusion sections the authors say: “In general, from fractal analysis, we found positive correlations or persistent behavior in brain activity to increase with longer dolphin-assisted therapy duration, since H values were the highest whenever the child patients underwent therapy”. This conclusion is not demonstrated by any of the results shown in the current version of this manuscript (in any figure or table). Why do the authors claim that an increase in the duration of the DTA could increase the H-values? Please show those results! because that would give much greater relevance to this study.

[Q4] If a simulation by mean of artificial generated biosignals (based on raw EEG recordings) from the BCI system is carried out, how could be obtained values of H > 0.5 during the DAT that ensure positive correlations (persistent data behavior) and therefore notable percentages of increases of H from Before to During DAT?

MINOR POINTS

[P1] Abstract (line 6); Introduction (line 105); and Conclusion (line 500): “TGAM1 –that is: ThinkGear-AM1 series of NeuroSky company ” instead of “TAGM1”.

[P2]The figure legends (boxes inside the figures with the different markers), the title of each axes, and also the scales of the axes do not look properly. You have to constantly make zoom to see the legends and scales. Authors should increase the size of these items in Figs. 1, 4, 6 and 8.

Reviewer 2 Report

This article is about monitoring the brain activity of patients with ISCP using Dolphin-Assisted Therapy (DAT). They have used a light-weight EEG sensor to monitor the patient's brain activity before and during DAT. The article has well written and understandable. 

However, there are some concern which should be addressed:

  • It would be good to apply noise reduction techniques as this sensor is quite noisy. Please refer to the following paper for noise reduction techniques (butterworth filtering and ICA technique) and get purer signal.

Nakisa, B., Rastgoo, M. N., Tjondronegoro, D. & Chandran, V. Evolutionary computation algorithms for feature selection of EEG-based emotion recognition using mobile sensors. Expert Syst. Appl. 93, 143–155 (2018)

  • it would be better to provide some literature review using this light-weight sensors in this domain and other domains. 
  • It is recommended to extract some other features and compare them, as we are not sure which features is performing better in classifying the changes before and during the test. 
